# Weibo-COV: A Large-Scale COVID-19 Social Media Dataset from Weibo

**Yong Hu[†], Heyan Huang[†], Anfan Chen[‡], Xian-Ling Mao[†]**
[†]Beijing Institute of Technology
{huyong,hhy63,maoxl}@bit.edu.cn
[‡]University of Science and Technology of China
caf16@ustc.edu.cn

## Abstract

With the rapid development of COVID-19 around the world, people are requested to maintain "social distance" and "stay at home". In this scenario, extensive social interactions transfer to cyberspace, especially on social media platforms like Twitter and Sina Weibo. People generate posts to share information, express opinions and seek help during the pandemic outbreak, and these kinds of data on social media are valuable for studies to prevent COVID-19 transmissions, such as early warning and outbreaks detection. Therefore, in this paper, we release a novel and fine-grained large-scale COVID-19 social media dataset collected from Sina Weibo, named Weibo-COV[1], contains more than 40 million posts ranging from December 1, 2019 to April 30, 2020. Moreover, this dataset includes comprehensive information nuggets like post-level information, interactive information, location information, and repost network. We hope this dataset can promote studies of COVID-19 from multiple perspectives and enable better and rapid researches to suppress the spread of this pandemic.

## 1 Introduction

At the beginning of this writing, COVID-19, an infectious disease caused by a coronavirus discovered in December, 2019, which also known as Severe Acute Respiratory Syndrome Coronavirus 2 (SARS-CoV-2), has caused 4,517,399 individuals infected globally, with a death toll of 308,515 (Doctor, 2020). Under the circumstance, the physical aspects of connection and human communication outside the household among people are limited considerably and mainly depend on digital devices like mobile phones or laptop computers (Abdul-Mageed et al., 2020). Due to it, people keep staying at home and spending more time on social media communication, making social media a vital avenue for information sharing, opinions expression, and help-seeking (Lopez et al., 2020). All that makes social media platforms like Weibo, Twitter, Facebook and Youtube a more vital sources of information during the pandemic.

In previous studies, social media was considered a valuable data source for research against disease, like uncovering the dynamics of an emerging outbreak (Zhang and Centola, 2019), predicting the flu activity, and disease surveillance (Jeremy et al., 2009). For example, some studies facilitate better influenza surveillance, like early warning and outbreaks detection (Kostkova et al., 2014; De Quincey and Kostkova, 2009), forecasting estimates of influenza activity (Santillana et al., 2015), and predicting the actual number of infected cases (Lampos and Cristianini, 2010; Szomszor et al., 2010). Hence, it is necessary to retrieve the relevant social media datasets and make it freely accessible for researchers, for the sake of public goods and facilitating the relevant studies of COVID-19.

In this paper, we release a novel large-scale COVID-19 social media dataset from Sina Weibo (akin to Twitter), one of the most popular Chinese social media platforms in China. For convenience, we named it Weibo-COV, which contains more than 40 million posts from December 1, 2019 to April 30, 2020. Specifically, unlike the traditional API-based data collection methods, which limit large-scale data access, in this study, we construct a high-quality Weibo active user pool with 20 million active users from over 250 million users, then collect all active users' posts during that period, followed by filtering COVID-19 related posts with 179 representative keywords. Moreover, the fields of posts in the dataset are fine-grained, including post-level information, interactive information, location information and repost network, etc. We

---

[1]https://github.com/nghuyong/weibo-public-opinion-datasets

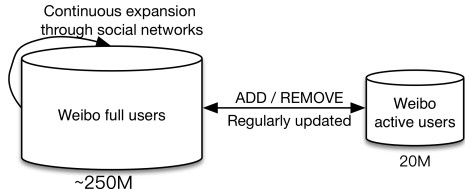

Figure 1: The construction of Weibo active user pool

hope this dataset can facilitate studies of COVID-19 from multiple perspectives and enable better and rapid research to suppress the spread of this disease.

## 2  Data Collection

### 2.1  Collection Strategy

At present, given specified representative keywords and a specified period, there are two kinds of methods for constructing Weibo post datasets: (1) Advanced searching API given by Weibo; (2) Traversing all Weibo users, collecting all their posts during the specified period, and then filtering these posts with specified keywords.

However, due to the limitation of the Weibo search API, the first method limits keyword search output to 50 pages (around 1000 posts), making it difficult to build large-scale datasets. As for the second kinds of method, although we could build large-scale datasets with almost no omissions, traversing all billions of Weibo users requires a very long time and massive bandwidth resources. Besides, a large proportion of Weibo users are inactive who may not post any posts in the specified period, and it makes meaningless to traverse their homepages.

To overcome these limitations, we propose a novel method to construct Weibo post datasets, which can build large-scale datasets with high construction efficiency. Specifically, we first build and dynamically maintain a high-quilty Weibo active user pool (just a small part of all users), and then we only traverse the home pages of these users and collect all their posts with specified keywords in a required period.

### 2.2  Weibo Active User Pool

As shown in Figure 1, based on initial seed users and continuous expansion through social relationships, we first collect more than 250 million Weibo users. Then we define that Weibo active users should meet the following two requirements: (1) The number of followers, fans and posts are all

Table 1: The field description of the dataset

| Field | Description |
|---|---|
| `_id` | the unique identifier of the post |
| `crawl_time` | crawling time of the post, which indicates when we retrieve the specific post from Weibo (GMT+8) |
| `created_at` | creating time of the post (GMT+8) |
| `like_num` | the number of like at the crawling time |
| `repost_num` | the number of repost at the crawling time |
| `comment_num` | the number of comment at the crawling time |
| `content` | the content of the post |
| `origin_weibo` | the `_id` of the origin post, only not empty when the post is a repost one |
| `geo_info` | information of latitude and longitude, only not empty when the post contains the location information |

more than 50; (2) The latest post is posted in 30 days. Therefore, we can build and dynamically maintain a Weibo active user pool from all collected Weibo users. Finally, the constructed Weibo active user pool contains 20 million users, accounting for 8% of the total number of Weibo users.

### 2.3  COVID-19 posts Collection

According to the collection strategies described in Section 2.1, we set the period from 00:00 December 1, 2019 (GMT+8, the date of the first confirmed infected case of COVID-19) to 23:59 April 30, 2020 (GMT+8). Following best practices of text retrieval and content analysis (Chen et al., 2019; Zhang et al., 2020; Li et al., 2020, 2019; Shen et al., 2020; Lacy et al., 2015), we generate a list of 179 keywords related to COVID-19 through close observation of Weibo posts every day from late January to April, 2020. These keywords are comprehensive, covering related terms such as coronavirus and pneumonia, as well as specific locations (e.g., "Wuhan"), drugs (e.g., "remdesivir"), preventive measures (e.g., "mask"), experts and doctors (e.g., "Zhong Nanshan"), government policy (e.g, "postpone the reopening of school") and others (see

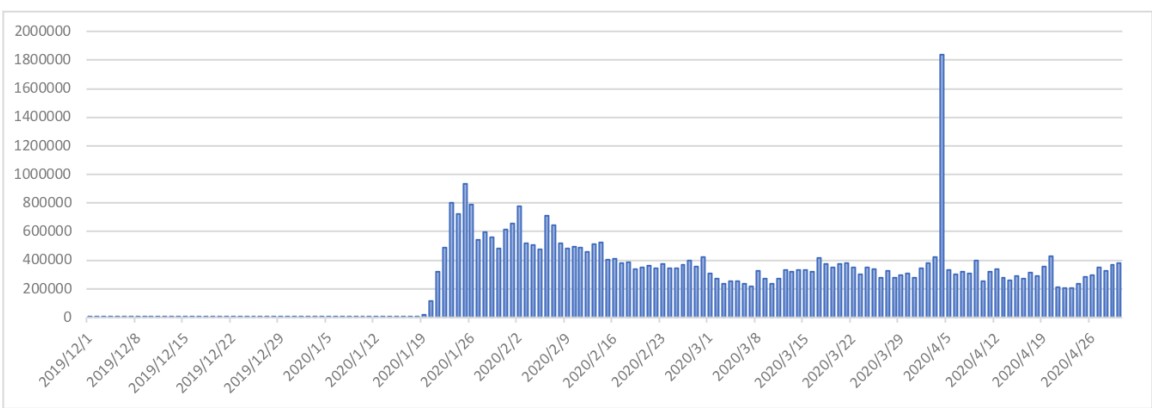

Figure 2: The daily distribution of Weibo-COV

Appendix.1 for the complete list).

As a result, based on 20 million Weibo active user pool, we first collect a total of 692,792,816 posts posted by these users in the specified period. Subsequently, we filter these posts by 179 keywords, along with duplication by unique post id. Finally, 40,893,953 posts are retained in our dataset.

Besides, some points should be noted that: (1) This COV-Weibo dataset can be retrieved with a single full download from our released website after submitting a data use agreement (DUA). (2) All the users' identifiable information such as user id, user name, post id, etc. have been converted into an unrecognizable status and can not be traced to protect the privacy of individual users, which is consistent with the presence of personally identifiable information (PII). Also, it is conducted by the terms-of-use of Weibo. (3) We declare the ownership of the source data to the corresponding Weibo users because Weibo users created this kind of public UGC (User Generated Content), and we only collect, organize and filter them. (4) Our Institutional Review Board (IRB) is under processing and waiting to be signed.

## 3 Data Properties

### 3.1 The Inner Structure of the Dataset

As shown in Table 1, fields of posts in the dataset are very rich, covering the basic information (`_id`, `crawl_time`, `content`), interactive information (`like_num`, `repost_num`, `comment_num`), location information (`geo_info`) and repost network (`origin_weibo`). Therefore, various kind of studies related to infectious diseases can be conducted based on this dataset, such as the impact

Table 2: The basic statistics of Weibo-COV

| #ALL | #GEO | #Original |
|------|------|-----------|
| 40,893,953 | 1,119,608 | 8,284,992 |

on people's daily life, the early characteristics of the disease, and government anti-epidemic policies.

### 3.2 Basic Statistic

As shown in Table 2, Weibo-COV contains a total number of 40,893,953 posts. Among these posts, there are 1,119,608 posts with geographic location information (accounting for 2.7%) and 8,284,992 original posts (accounting for 20.26%).

### 3.3 Daily Distribution

The distribution of the number of posts by day is shown in Figure 2. It can be noticed that from December 1, 2019 to January 18, 2020, the number of COVID-19 related posts is tiny (less than 10K) and may include some noise data. Since January 19, 2020, the number of COVID-19 related posts expanded rapidly and maintained at least 200,000 per day.

Note that the data on April 4, 2020, is particularly striking, and the number of posts on that day exceeds 1.8 million. A reasonable explanation could be that day was Chinese Tomb Sweeping Festival, a national mourning was held for the compatriots who died in the epidemic, people posted or reposted many mourning posts on Weibo on that day, which drawn extensive attention and generated a massive number of posts.

### 3.4 GEO Distribution

As shown in Figure 3, we plot the location distribution of posts with geoinformation on April 4,

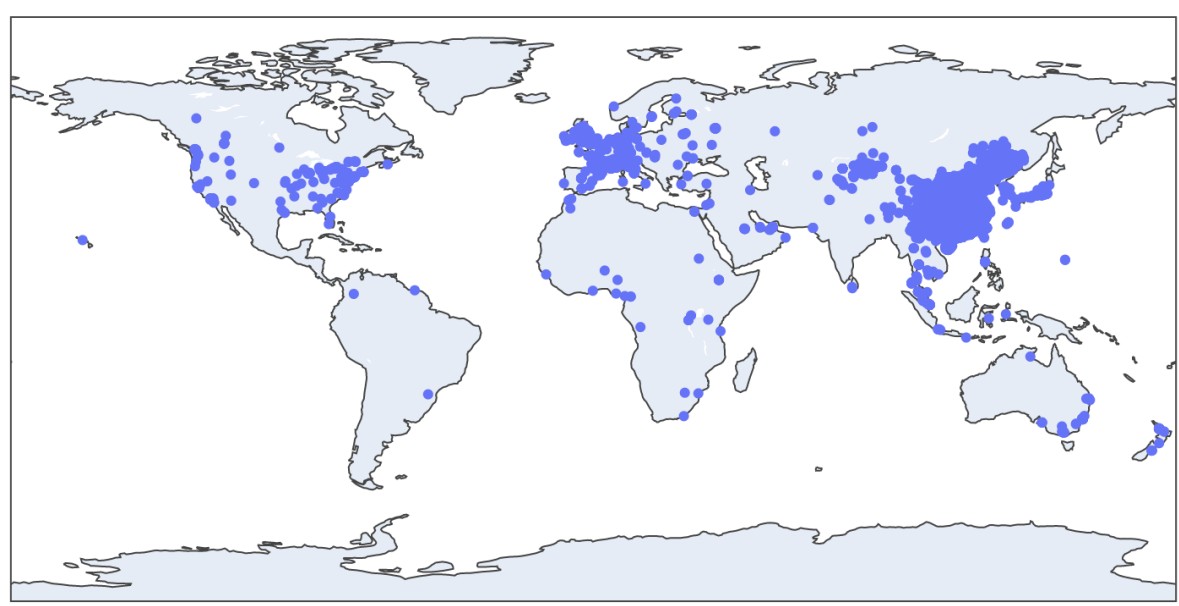

Figure 3: Distribution of location information of posts on April 4, 2020

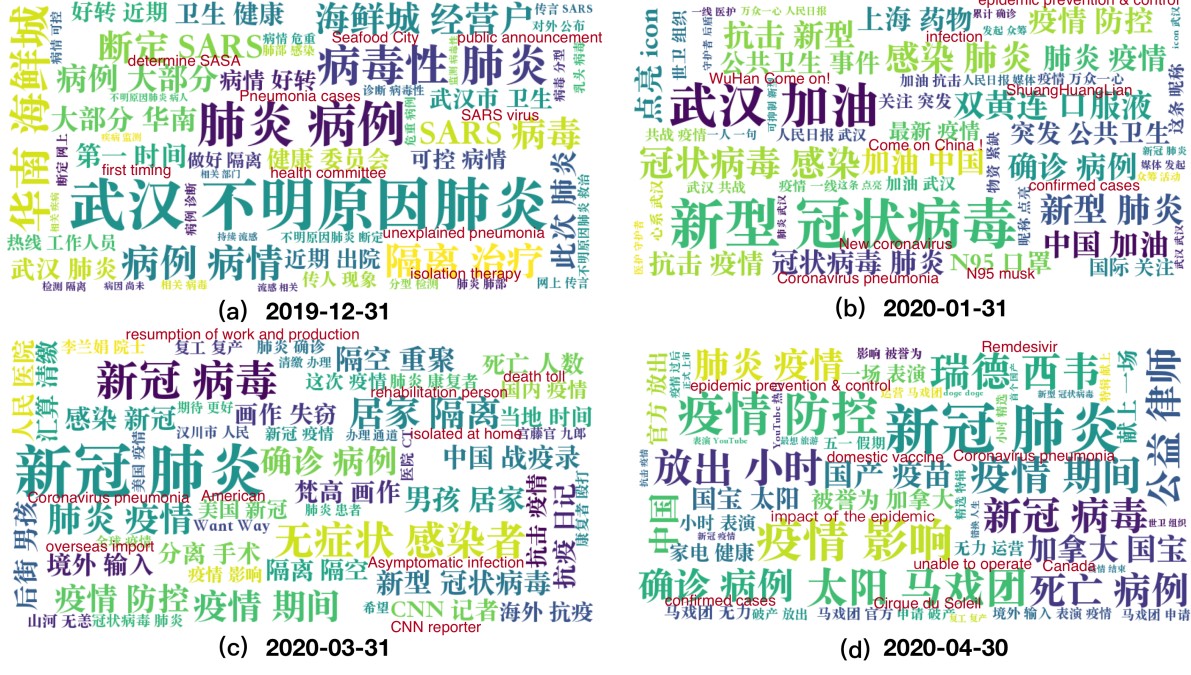

Figure 4: Word cloud of posts in four days and some words are translated in red

2020. It can be seen that the distribution of posts is mainly in China. There is also a part of posts distributed oversea, including major countries in Asia, Europe, Australia and America. The possible reasons could be that with the development of economic globalization, more and more Chinese people go abroad for work/living, and more and more foreigners start to use Weibo, promoting a large proportion of oversea Weibo users.

Therefore, our dataset can provide insight into the nationwide and global impact of the pandemic.

## 3.5 Word Cloud

We select four days of posts data at different stages of the epidemic development and draw word clouds. As shown in Figure 4 (a), in the early days, people even did not know the characteristics of the virus and, however, the government began to take preliminary actions (e.g., "unexplained pneumonia" and "health committee"). Later, as shown in Figure 4 (b), people learned that the virus is a new coronavirus and learned preventive methods and

medicines (e.g., "new coronavirus", "N95 mask" and "ShuangHuang Lian"). Then, as shown in Figure 4 (c), governments took strict isolation rules and strove to prevent imported cases from abroad (e.g., "isolated at home" and "overseas import"). By the end of April, as shown in Figure 4 (d), the virus has had many impacts on people's lives. Fortunately, research on vaccines and medicines has been ongoing and made significant progress (e.g., "Remdesivir").

Therefore, this dataset runs through the whole development stages of COVID-19, including impacts of the disease on all aspects of society.

## 4 Related Work

Several works have focused on creating social media datasets for enabling COVID-19 research. (Chen et al., 2020), (Lopez et al., 2020) and (Abdul-Mageed et al., 2020) have already released datasets collected from Twitter. However, these datasets are mainly in English, and posts generated by Chinese, the epicentre of the early development of COVID-19, also deserves close attention. Therefore, collecting the Weibo datasets are also valuable and can provide additional supplements for researches.

Only one dataset proposed by (Gao et al., 2020) includes posts from Weibo, but their method relies on Weibo advanced search API provided by Weibo, which hinders them from collecting large-scale posts as we mentioned above. Compared with our dataset, the data size (less than 200K), the time period (from January 20, 2020 to March 24, 2020), and the number of keywords (only four keywords) of this Weibo dataset seem much smaller and narrow.

## 5 Conclusion

In this paper, we release Weibo-COV, a first large-scale COVID-19 posts dataset from Weibo. The dataset contains more than 40 million posts from December 1, 2019 to April 30, 2020, with rich field information. We hope this dataset could promote and facilitate related studies on COVID-19.

## 6 Acknowledgments

We would like to thank all the reviewers for their helpful suggestions and comments. This work is supported by the National Key R&D Plan (No. 2016QY03D0602), NSFC (No. U19B2020, 61772076, 61751201 and 61602197), NSFB (No. Z181100008918002), the 25th department funding of USTC (No. DA2110251001) and 2019 New Humanities Funding of USTC (No. YD2110002015).

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

# A Appendices

## A.1 Covid-19 Related Keywords

Table 3: The list of selected keywords related to COVID-19

| Keywords | Translations |
|---|---|
| 冠状 | Coronavirus |
| Cov-19 | Cov-19 |
| 新冠 | Coronavirus |
| 感染人数 | Infected cases |
| N95 | N95 Mask |
| 大众畜牧野味店 | Dazhong wildlife restaurant |
| 华南野生市场 | South China wild market |
| 管轶 | Guan Yi |
| 武汉病毒所 | Wuhan Institute of Virology |
| CDC | Center for Disease Control and Prevention |
| 中国疾病预防控制中心 | Chinese Center for Disease Control and Prevention |
| 疾控中心 | Center for Disease Control and Prevention |
| #2019nCoV | #2019nCoV |
| 双黄连 AND 抢购 | Shuanghuanglian AND Rush to buy |
| 双黄连 AND 售罄 | Shuanghuanglian AND Sold out |
| 武汉卫健委 | Wuhan Municipal Health Committee |
| 湖北卫健委 | Health Commission of Hubei Province |
| #nCoV | #nCoV |
| PHEIC | PHEIC |
| 疫情 | Epidemic outbreak |
| 火神山 | Huoshen Shan hospital |
| 雷神山 | Leishen Shan hospital |
| 钟南山 | Zhong Nanshan |
| Coronavirus | Coronavirus |
| Remdesivir | Remdesivir |
| 瑞德西韦 | Remdesivir |
| 感染 AND 例 | Infected AND cases |
| 武汉 AND 封城 | Wuhan AND Lockdown |
| 高福 | George Fu Gao |
| 王延轶 | Wang Yanyi |
| 舒红兵 | Shu Hongbing |
| 协和医院 | Xiehe Hospital |
| 武汉 AND 隔离 | Wuhan AND Quarantine |
| 李文亮 AND 医生 | Doctor AND Li Wenliang |
| 云监工 | Supervising work on cloud |
| 武汉仁爱医院 | Wuhan Ren'ai Hospital |
| 黄冈 AND 感染者 | Huanggang AND Infected cases |
| 孝感 AND 感染者 | Xiaogan AND Infected cases |
| 居家隔离 | Isolated at home |
| 防护服 | Protective Clothing |
| 隔离14天 | Isolation AND 14 days |
| 潜伏期 AND 24天 | Incubation period AND 24 days |
| 潜伏期 AND 14天 | Incubation period AND 14 days |
| 国际公共卫生紧急事件 | International Public Health Emergencies |

**Table 3 – continued from previous page**

| Keywords | Translations |
|---|---|
| 方舱医院 AND 武汉 | FangCang Hospital AND Wuhan |
| 一省包一市 | one province gives a hand to one Hubei city |
| 晋江毒王 | Super spreader of COVID-19 in Jinjiang |
| 超级传播者 | Super spreader |
| 湖北 AND 王晓东 | Hubei AND Wang Xiaodong |
| 蒋超良 | Jiang Chaoliang |
| 李文亮 | Li Wenliang |
| 千里投毒 | Spread Virus from a thousand miles |
| 武汉病毒研究 | Virology research in Wuhan |
| 武汉 AND 李医生 | Wuhan AND Li Wenliang |
| 国家疾控中心 | Chinese Center for Disease Control and Prevention |
| 武汉 AND 疫苗 | Wuhan AND Vaccine |
| 武汉 AND 征用宿舍 | Wuhan AND Requisitioned students' dormitory |
| 周佩仪 | Zhou Peiyi |
| 武汉中心医院 | The Central Hospital of Wuhan |
| 张晋 AND 卫健委 | Zhang Jin AND Health Commission |
| 张晋 AND 卫生将康委员会 | Zhang Jin AND Health Commission |
| 刘英姿 AND 卫健委 | Liu Yingzi AND Health Commission |
| 刘英姿 AND 卫生健康委员会 | Liu Yingzi AND Health Commission |
| 王贺胜 AND 卫健委 | Wang Hesheng AND Health Commission |
| 王贺胜 AND 卫生健康委员会 | Wang Hesheng AND Health Commission |
| 复工 | Enterprise work resuming |
| 中小企业 AND 困境 | Small and medium-sized enterprise AND Dilemma |
| 武汉 AND 死亡病例 | Wuhan AND Death cases |
| 武汉 AND 感染病例 | Wuhan AND Infection cases |
| 湖北 AND 死亡病例 | Hubei AND Death cases |
| 湖北 AND 感染病例 | Hubei AND Infected cases |
| 中国 AND 死亡病例 | China AND Death cases |
| 中国 AND 感染病例 | China AND Infected cases |
| 潜伏期 | Incubation Period |
| 北京 AND 病例 | Beijing AND Cases |
| 天津 AND 病例 | Tianjin AND Cases |
| 河北 AND 病例 | Hebei AND Cases |
| 辽宁 AND 病例 | Liaoning AND Cases |
| 上海 AND 病例 | Shanghai AND Cases |
| 江苏 AND 病例 | Jiangsu AND Cases |
| 浙江 AND 病例 | Zhejiang AND Cases |
| 福建 AND 病例 | Fujian AND Cases |
| 山东 AND 病例 | Shandong AND Cases |
| 广东 AND 病例 | Guangdong AND Cases |
| 海南 AND 病例 | Hainan AND Cases |
| 山西 AND 病例 | Shanxi AND Cases |
| 内蒙古 AND 病例 | Inner Mongolia AND Cases |
| 吉林 AND 病例 | Jilin AND Cases |
| 黑龙江 AND 病例 | Heilongjiang AND Cases |
| 安徽 AND 病例 | Anhui AND Cases |
| 江西 AND 病例 | Jiangxi AND Cases |
| 河南 AND 病例 | Henan AND Cases |

| Keywords | Translations |
|---|---|
| 湖北 AND 病例 | Hubei AND Cases |
| 湖南 AND 病例 | Hunan AND Cases |
| 广西 AND 病例 | Guangxi AND Cases |
| 四川 AND 病例 | Sichuan AND Cases |
| 贵州 AND 病例 | Guizhou AND Cases |
| 云南 AND 病例 | Yunnan AND Cases |
| 西藏 AND 病例 | Tibet AND Cases |
| 陕西 AND 病例 | Shanxi AND Cases |
| 甘肃 AND 病例 | Gansu AND Cases |
| 青海 AND 病例 | Qinghai AND Cases |
| 宁夏 AND 病例 | Ningxia AND Cases |
| 新疆 AND 病例 | Xinjiang AND Cases |
| 香港 AND 病例 | Hong Kong AND Cases |
| 澳门 AND 病例 | Macau AND Cases |
| 台湾 AND 病例 | Taiwan AND Cases |
| ECOM | Extracorporeal Membrane Oxygenation |
| sars-cov-2 | sars-cov-2 |
| 复学 | Resumption of schooling |
| 护目镜 | Goggles |
| 核酸检测 | nucleic acid testing (NAT) |
| COVID-19 | COVID-19 |
| 2019-nCoV | 2019-nCoV |
| 疑似 AND 病例 | Suspicious cases |
| 无症状 | Asymptomatic Patients |
| 累计病例 | Cumulative confirmed cases |
| 境外输入 | imported cases of NCP |
| 累计治愈 | Cumulative cured cases |
| 绥芬河 | Sui Fenhe |
| 舒兰 | Shu Lan |
| 健康码 | Health QR code |
| 出入码 | Community Access Code |
| 返校 | Back to Camp |
| 美国 AND 例 | USA AND Cases |
| 西班牙 AND 例 | Spain AND Cases |
| 新加坡 AND 例 | Singapore AND Cases |
| 加拿大 AND 例 | Canada AND Cases |
| 英国 AND 例 | UK AND Cases |
| 印度 AND 例 | India AND Cases |
| 日本 AND 例 | Japan AND Casess |
| 韩国 AND 例 | South Korea AND Cases |
| 德国 AND 例 | Germany AND Cases |
| 法国 AND 例 | France AND Cases |
| 意大利 AND 例 | Italy AND Cases |
| 新增 AND 例 | New AND Cases |
| 人工膜肺 | Extracorporeal Membrane Oxygenation |
| 双盲测试 | Double Blind Test |
| 疫苗 | Vaccine |
| 小区出入证 | Community Entry card |

| Keywords | Translations |
|---|---|
| 战疫 | Anti-COVID-19 |
| 抗疫 | Anti-COVID-19 |
| 湖北卫健委 AND 免职 | Health commission of Hubei Province AND Remove from the position |
| 发热患者 | Fever patients |
| 延迟开学 | Postpone the reopening of school |
| 开学时间 AND 不得早于 | The start time of school AND Not earlier than |
| 累计死亡数 | Cumulative deaths |
| 疑似病例 | Suspicious cases |
| 入户排查 | Household troubleshoot |
| 武汉 AND 肺炎 | Wuhan AND Pneumonia |
| 新型肺炎 | Novel Pneumonia |
| 不明原因肺炎 | Pneumonia of unknown cause |
| 野味肺炎 | Wildlife pneumonia |
| 出门 AND 戴口罩 | Going out AND Wear mask |
| 3M AND 口罩 | N95 AND Mask |
| KN95 AND 口罩 | 3M AND Mask |
| 新肺炎 | Novel Pneumonia |
| #2019nCoV | #2019nCoV |
| 新型肺炎 AND 死亡 | Novel Pneumonia AND Death |
| 新型肺炎 AND 感染 | Novel Pneumonia Infection |
| 武汉 AND 肺炎 AND 谣言 | Wuhan AND Pneumonia AND Rumors |
| 8名散布武汉肺炎谣言 | Eight people AND Spread rumors of Wuhan pneumonia |
| 黄冈 AND 新肺炎 | Huanggang AND Novel Pneumonia |
| 孝感 AND 新肺炎 | Xiaogan AND Novel Pneumonia |
| 居家隔离 | Isolated at home |
| 武汉中心医院 AND 新型肺炎 | The Central Hospital of Wuhan AND Novel Pneumonia |
| 武汉肺炎 | Wuhan Pneumonia |
| 企业复工 | Enterprise work resuming |
| 囤积口罩 | Hoarding mask |
| 零号病人 | Zero Patient |
| 黄燕玲 | Huang Yanling |
| 病毒源头 | Oringin of Cov-19 |
| 电子烟肺炎 AND 新型冠状 | E-cigarette Pneumonia AND Coronavirus |
| 病毒战 | Virus War |
| 病毒 AND 实验室泄露 | Virus AND laboratory leakage |
| 比尔盖茨 AND 疫苗牟利 | Bill Gates AND Vaccine for profit |
| 美国细菌实验室 | US Army Bacterial Laboratory |
| 确诊 | Confired Infencted COV-19 cases |
| pandemic | pandemic |