# OpenReview forum: "Weibo-COV: A Large-Scale COVID-19 Social Media Dataset from Weibo"
_EMNLP/2020/Workshop/NLP-COVID — NLP-COVID19-EMNLP Poster_

### Official Review · AnonReviewer2 · 2020-09-19
**A useful resource paper**

**Rating:** 6
**Confidence:** 3

**Review:**

This paper describes a dataset for COVID-19 built from Weibo, a Chinese social media platform.
The authors first maintain a Weibo active user pool -- around 20 million active users, and then collect their tweets using a list of specified keywords relating to COVID-19 (179 in total).

Strengths：
- A large scale dataset, containing tweet content, interactive, location and retweet information. Not only text-based but also social network-based analysis can be potentially conducted on the dataset.
- The authors describe in detail how they build the dataset

Weaknesses:
- I am not sure whether collecting only active user's tweets makes sense. It is possible that some users create and start to post tweets because of (after) the pandemic. It may be worth analysing those users as well.

Questions and suggestions to authors:
- Are these keywords manually choosen or decided based on their popularity? Also it looks to me some keywords are repeated. For example, is the result filtered using keyword 'Doctor AND Li Wenliang' a strict subset of result using keyword 'Li Wenliang'?
- The title of (Chen et al, 2020) paper has been updated to 'Tracking Social Media Discourse About the COVID-19 Pandemic: Development of a Public Coronavirus Twitter Data Set'

---

> ### Author Response · Authors · 2020-09-27
> **Detailed Response**
>
> **I am not sure whether collecting only active user's tweets makes sense. It is possible that some users create and start to post tweets because of (after) the pandemic. It may be worth analysing those users as well.**
>
> Thank you for this suggestion. These sounds make sense to including the medium active users in our dataset, such as those newly created accounts and start to post tweets because of (after) the pandemic. We will update our dataset according to this valuable advice.
>
> **Are these keywords manually choosen or decided based on their popularity? Also it looks to me some keywords are repeated. For example, is the result filtered using keyword 'Doctor AND Li Wenliang' a strict subset of result using keyword 'Li Wenliang'?**
>
> I appreciate this suggestion. Following best practices for text retrieval and analysis(Lacy, 2015), we generated a comprehensive list of keywords related to COVID-19 through close observation of Weibo posts every day from late January to March 2020. And, the repeated keywords combination is not a severe problem because we have removed duplicate posts by post id at the following stage. We apologize for not indicating the deduplication procedure in our main text. We will add a sentence to indicate it in our new version.
>
> Lacy S, Watson BR, Riffe D, Lovejoy J. Issues and Best Practices in Content Analysis. Journal Mass Commun Q 2015 Sep 28;92(4):791-811.
>
> **The title of (Chen et al, 2020) paper has been updated to 'Tracking Social Media Discourse About the COVID-19 Pandemic: Development of a Public Coronavirus Twitter Data Set'**
>
> Thank you for this suggestion, we will update this reference to be more accurate.

---

### Official Review · AnonReviewer3 · 2020-09-21
**A short paper introducing a collection of covid-19-related tweets from Weibo**

**Rating:** 6
**Confidence:** 3

**Review:**

This is a short paper providing an overview of the collection of tweets posted on a popular Chinese social media platform called Weibo from December 1, 2019 to April 30, 2020. The authors first create a pool of active users by filtering them based on their activity and the number of tweets, followers, and fans. They then compile a collection of tweets from these users using 179 manually pre-defined keywords.
Several questions and comments:
- Why was the data collected only until the end of April? The authors claim that the pool of active users is dynamically maintained and if so it would be better to provide both the users and the readers with a more recent version of the dataset.
- In what languages are the tweets collected in Weibo-COV? I assume the majority is in Chinese, this should be stated explicitly somewhere at the beginning of the paper.
- What does crawling time mean in this context? Please either specify in Table 1 or add the definition to the main text.
- Not sure what is meant by “Cirque du Soleil in Canada” in section 3.5, please review.
- Typos and grammar mistakes occur throughout the text (e.g. “N95 musk”). It would be nice to correct those to improve the readability of the paper.

---

> ### Author Response · Authors · 2020-09-27
> **Detailed Response**
>
> **Why was the data collected only until the end of April? The authors claim that the pool of active users is dynamically maintained and if so it would be better to provide both the users and the readers with a more recent version of the dataset.**
>
> Thank you for this advice. We collected the COVID-19 related posts until the end of April for there are quite a few newly confirmed infected cases relatively at this stage, which indicates the pandemic is under control in China. However, many scholars are requesting us to provide a recent version of the dataset, so we are preparing to update our dataset and make it accessible to the researchers.
>
> **In what languages are the tweets collected in Weibo-COV? I assume the majority is in Chinese, this should be stated explicitly somewhere at the beginning of the paper.**
>
> Yes, most of the keywords are in Chinese, some are in English like “COVID-19”, “#2019nCoV”, and “#nCoV” et al. We will emphasize this point in the new version of our paper.
>
> **What does crawling time mean in this context? Please either specify in Table 1 or add the definition to the main text.**
>
> Thank you very much for this gentle reminder. The crawling time of the tweet means when we retrieve the specific post from Weibo. We will specify it in the main text accordingly.
>
>
> **Not sure what is meant by “Cirque du Soleil in Canada” in section 3.5, please review.**
> **Typos and grammar mistakes occur throughout the text (e.g. “N95 musk”). It would be nice to correct those to improve the readability of the paper.**
>
> Thanks for your suggestion. “Cirque du Soleil in Canada” is a specific entity in Canada, we will translate it in English in the new version of our paper. I am so sorry for the typos and grammar mistakes, and will carefully recheck our paper to improve the readability of the paper.

---

### Official Review · AnonReviewer1 · 2020-09-23
**A large-scale dataset that may be useful for further studies on COVID-19**

**Rating:** 5
**Confidence:** 3

**Review:**

The paper introduces a large-scale dataset with posts related to COVID-19 collected from Weibo, a social media in China. It describes in detail the process to collect the dataset and provides some interesting analysis of the dataset.

Pros:
* The dataset  is large and contains rich information such as location and post time. This could be helpful for further COVID-related researches.
* This paper provides interesting analysis about the dataset such as the daily distribution of COVID-related posts and word clouds at different phases.

Cons:
* There are no deduplication steps to deduplicate tweets in the dataset. The keyword-based collection could result in many duplicated tweets, for example, the same announcements from the government reported by lots of different news accounts, and popular tweets retweeted by other accounts. The duplicated tweets could bias the statistics of the dataset and further studies.
* While the main contribution of this paper is a large-scale dataset, there is no direct evaluation of its quality. It is unknown how many tweets in this dataset are directly related to COVID-19. For example, some keywords (e.g. person and location) could potentially pick up non-COVID-related posts (e.g. searching "Li Wenliang" could pick up tweets that falsely accused him of spreading rumors or tweets that described his profile).  Another example is that a portion of words from Fig.4c and Fig.4d are not directly related to COVID-19, such as "汇算 清缴"(settlement and payment), "梵高 画作"(paintings of Vincent Van Goph) and ”后街男孩“(Backstreet Boys).  The precision of this keyword-based approach to collect COVID-related tweets remains unknown, which makes the dataset less useful to the community.

Suggestion:
* Some Chinese keywords do not correctly match their translation in English in the appendix table .  For example, "美国 AND 例" should be translated to "USA AND cases " instead of "USA COV-19 AND cases".

---

> ### Author Response · Authors · 2020-09-27
> **Detailed Response**
>
> **There are no deduplication steps to deduplicate tweets in the dataset. The keyword-based collection could result in many duplicated tweets, for example, the same announcements from the government reported by lots of different news accounts, and popular tweets retweeted by other accounts. The duplicated tweets could bias the statistics of the dataset and further studies.**
>
> We really like this suggestion. However, the repeated keywords combination is not a severe problem because we have removed duplicate posts by post id at the following stage. We apologize for not indicating the deduplication procedure in our main text. We will add a sentence to indicate it in our new version.
>
> **While the main contribution of this paper is a large-scale dataset, there is no direct evaluation of its quality. It is unknown how many tweets in this dataset are directly related to COVID-19. For example, some keywords (e.g. person and location) could potentially pick up non-COVID-related posts (e.g. searching "Li Wenliang" could pick up tweets that falsely accused him of spreading rumors or tweets that described his profile). Another example is that a portion of words from Fig.4c and Fig.4d are not directly related to COVID-19, such as "汇算 清缴"(settlement and payment), "梵高 画作"(paintings of Vincent Van Goph) and ”后街男孩“(Backstreet Boys). The precision of this keyword-based approach to collect COVID-related tweets remains unknown, which makes the dataset less useful to the community.**
>
> We highly appreciate the reviewer’s suggestion about the noisy posts. We will narrow the scope of our keywords combination afterward to filter out the irrelevant posts from the current dataset, which may not undermine the completeness of our COVID-19 Weibo dataset. I think this is doable because we already have a bigger corpus.
>
> **Some Chinese keywords do not correctly match their translation in English in the appendix table . For example, "美国 AND 例" should be translated to "USA AND cases " instead of "USA COV-19 AND cases".**
>
> Thank you for carefully reading our paper. We are modifying the translation table carefully according to your suggestion.

---

### Comment · Program_Chairs · 2020-10-02
**Data release characteristics need clarification in manuscript**

I would firmly recommend that the authors address the following aspects in the manuscript:
In what manner the data is available -- full single download, or 'tweet'-ids; the requirement for researchers to sign a data use agreement; ownership of the source data (Weibo, or its users); how this data dissemination effort is compatible with the Weibo terms-of-use; presence of personally identifiable information (PII) such as user names in the data, and efforts undertaken (yes/no) to remove these; whether or not ethics board approval was sought / granted.

As a minor comment: the term 'tweet' seems proprietary to Twitter. Is there not a better name for Weibo messages?
Also: several references are incomplete or incorrectly formatted (e.g. ginsberg et al, nature)